# A DC Microgrid System for Powering Remote Areas †

**Tri Ardriani [1], Pekik Argo Dahono [1,*], Arwindra Rizqiawan [1], Erna Garnia [2], Pungky Dwi Sastya [3], Ahmad Husnan Arofat [3] and Muhammad Ridwan [3]**

1. Institut Teknologi Bandung, School of Electrical Engineering and Informatics, Jl. Ganesha 10, Bandung 40111, Indonesia; ardriani.t@gmail.com (T.A.); windra@std.stei.itb.ac.id (A.R.)
2. Faculty of Economics, Universitas Sangga Buana, Jl. PHH Mustofa (Suci) 68, Bandung 40111, Indonesia; erna.garnia@usbypkp.ac.id
3. Technology Development Division, PT. Len Industri (Persero), Jl. Soekarno-Hatta 442, Bandung 40111, Indonesia; pungkydwisastya@gmail.com (P.D.S.); ahmad.husnan@len.co.id (A.H.A.); muhammad.ridwan@len.co.id (M.R.)
* Correspondence: pekik@konversi.ee.itb.ac.id
† This paper is an extended version of our paper published in the Proceeding of 2018 Conference on Power Engineering and Renewable Energy (ICPERE), 29–31 October 2018, Solo, Indonesia.

**Abstract:** DC microgrid has been gaining popularity as solution as a more efficient and simpler power system especially for remote areas, where the main grid has yet to be built. This paper proposes a DC microgrid system based on renewable energy sources that employs decentralized control and without communication between one grid point and another. It can be deployed as an individual isolated unit or to form an expandable DC microgrid through DC bus for better reliability and efficiency. The key element of the proposed system is the power conditioner system (PCS) that works as an interface between energy sources, storage system, and load. PCS consists of modular power electronics devices and a power management unit, which controls power delivery to the AC load and the grid as well as the storage system charging and discharging sequence. Prototypes with 3 kWp solar PV and 13.8 kWh energy storage were developed and adopt a pole-mounted structure for ease of transportation and installation that are important in remote areas. This paper presents measurement results under several conditions of the developed prototypes. The evaluation shows promising results and a solid basis for electrification in remote areas.

**Keywords:** DC microgrid; power conditioner system; renewable energy; scalable microgrid





## 1. Introduction

As an archipelago with more than 17,000 islands, Indonesia faces a challenge in delivering electricity to all its citizens, particularly to those who live in the remote areas and outer islands. According to the Indonesian Ministry of Energy and Mineral Resources [1], although the total electrification ratio of Indonesia in 2019 has reached almost 99%, there are places that lag behind and around 1 million families still without access to electricity. Additionally, strong grids are only available to the main islands of the country, where the central government and most of the population live. There are also areas where electricity is available only for several hours a day.

Located in the equator, the solar potential in Indonesia is estimated to be around 208 GWp, much higher than other types of RES such as hydro (75 GWp), wind (60 GWp), and geothermal (29.5 GWp) [2]. It is one of the most evenly distributed RES throughout the country. Therefore, a solar-based system is very suitable to accelerate providing electricity to rural villages.

Solar PV has been a popular choice RES because they are getting cheaper by the day and are easy to install. In urban cities, small-scale solar power systems are installed on rooftops, such as described in [3,4], both to provide green energy and to reduce bills. The disadvantage of PV rooftops is that it usually needs the AC grid to run and cannot

operate in stand-alone mode. Extending the main grid to remote areas require a lot of time and effort. One viable solution to this problem is to build independent power systems that do not need to rely on the main grid, i.e., a microgrid. These systems tap into RES near the local load, effectively eliminating the cost needed to draw long cables from the main grid and reducing dependency on fossil fuel-based power generation.

There are two types of microgrids, AC and DC microgrids. In AC microgrids, energy sources that produce DC power, such as PV panels and a fuel cell, will obviously need DC-AC conversion to connect to the lines. Interestingly, sources that produce AC power, such as wind, hydro, and geothermal, may require AC-DC-AC conversion for better synchronized connection to the grid [5–7]. Meanwhile, in DC microgrid, both DC-producing and AC-producing sources may require only one conversion, resulting in fewer converters needed, which in turn gives better efficiency. Nowadays, DC microgrid is gaining popularity due to its simplicity and higher power quality than its AC counterpart [7–9]. In DC system, the control is simpler because there is no problem with synchronization and reactive power [5–17].

Many DC microgrid systems have been proposed in the literature. Refs. [18,19] discuss microgrid systems that are reliant on AC utility. These systems are similar to the ones in [3,4] in that they are more suited for urban areas where the network is strong. In [10], a DC microgrid system for rural areas is designed to be able to operate independently in the absence of power network, but it only has stand-alone mode, meaning it does not have power sharing capability.

For rural areas, building centralized microgrids is a poor choice [11–13]. This is due to the socio-environmental conditions of rural areas. Firstly, the geographical terrain is difficult. This calls for a system that can be easily transported and installed. Secondly, communities are formed in clusters where homes, schools, hospital, and other public facilities are built apart from each other, which means a centralized generation system will have a lot of conductor losses. Thirdly, in rural areas, communication infrastructure is lacking. Therefore, the system needs to be able to operate in stand-alone mode. To accommodate future expansion, the equipment that go into the system have to be modular and can be installed in a plug-and-play manner [5].

Distributed DC microgrid systems are proposed in [11–17]. Refs. [11–15] take into account the fact that a lot of home appliances can run on low-voltage DC power. These devices also do not require a lot of power. Because of that, the system can be designed with low specification that costs considerably less than other systems. Unfortunately, its strength can also be its weakness, because at present, AC-powered home appliances are still very common.

The systems described in [16,17] are made of a DC bus and power converters that interface it with the energy source, ESS, and other microgrid elements. Loads will tap into the DC bus directly. This system can be built for large scale; however, loads may be located far away from the source and the distribution losses may be quite high.

In addition to the topology, discussion on power electronics technology and control methods is indispensable. There are a lot of different power converter topologies to choose from, all ranging from basic, uni-directional DC-DC converter, to multilevel and modified converters [14,16,20–23] with unique features that can be harnessed for a plethora of different purposes. Although, a system with many different kinds of power converters, such as in [10,16,24], may be expensive due to the higher design and production cost. Furthermore, replacement units may be limited to the same manufacturer, making it an inflexible system. Ref. [25] proposes a uniform design of multi-purpose converters for microgrids, but presently the technology has only been applied to DC-to-DC conversion.

In the realm of control techniques, one of the central issues is the power sharing method between microgrid elements. Ref. [26] proposes forecasting algorithm for an isolated DC microgrid system to regulate power flow. This algorithm has succeeded in reducing generation costs, but the predictive aspect of the control is prone to uncertainties and complex to implement. Ref. [27] propose droop control method based on state of

charge (SoC). These methods are easier to implement, especially because in rural areas, maintenance and operation have to be kept simple. Ref. [28] proposes using peer-to-peer control between microgrids, while [29] opts using switching frequency modulation-based communication, which is a good choice for systems in urban area where communication link can be established fairly easily, but not as feasible in the country and remote islands. Generally, individual microgrid control is preferred because failure in one system may not affect the others [30,31].

This paper proposes a DC microgrid system that has all the requirements mentioned above, as well as the main building block of the system. The key features of this system are as follow: (1) Modular; (2) expandable; (3) independent/without communication; (4) easily transported; and (5) each system covers a small ground area. The last point is crucial for places with land-ownership problems. Figure 1 shows a multi-point DC microgrid of the proposed system, with the primary equipment denoted as PCS. Each system consists of the following four elements: (1) A locally available energy source; (2) ESS; (3) AC loads; and (4) DC bus interconnection. If more points are to be integrated, they can be connected via the DC grid. Each point is capable of both taking and giving energy from and to the grid.

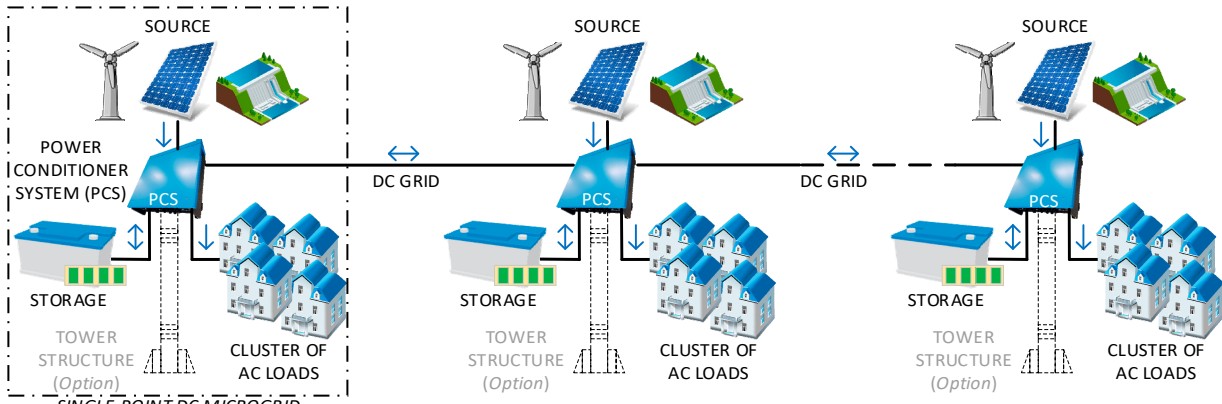

**Figure 1.** The proposed DC microgrid system.

This paper is structured as follows: PCS as the main element of the proposed system will be explained in Section 2. Comparison with other DC microgrid systems that have been proposed and implemented for rural areas will also be covered in this section. Section 3 discusses the evaluation of PCS to show the feasibility of implementing this system. Section 4 touches on the business and investment side of the proposed DC microgrid installation for a remote area in Indonesia. Finally, Section 5 contains the conclusion to this paper.

## 2. A Modular, Independent, and Expandable DC Microgrid System

### 2.1. Power Conditioner System

PCS is the primary element in the DC microgrid system proposed in this paper. It interfaces an energy source such as solar PV, wind, and hydro, an ESS, AC loads, and the DC grid. PCS consists of power converters to harvest energy from the source, charge and discharge the ESS, and regulate power to loads and grid. Figure 2a shows the inner configuration of a PCS. It may be deployed to form an isolated, independent system, but it can also connect to other PCS and share power between each other. Each PCS has its own management unit that keeps track the energy stored in its ESS, and export or import power to and from its neighboring points according to this data [32].

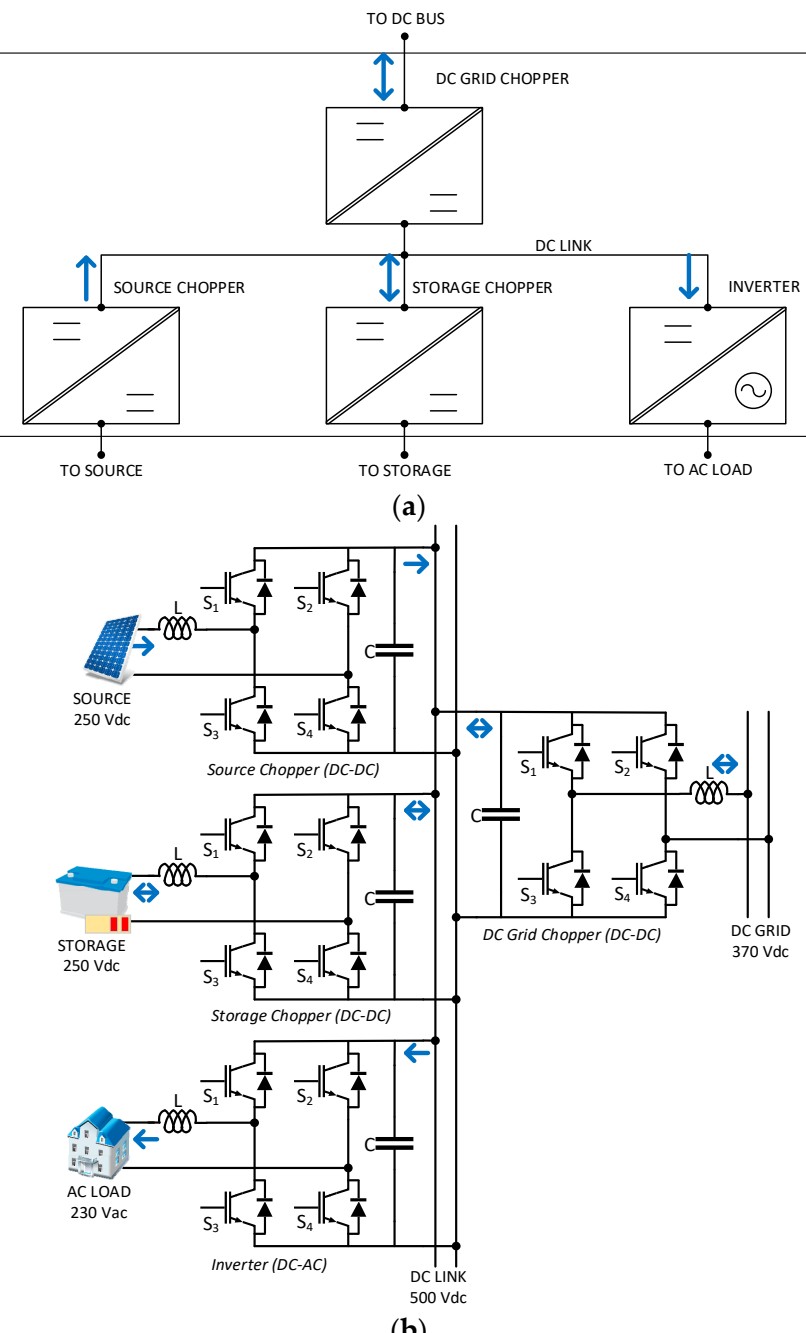

**Figure 2.** (**a**) Block diagram of a power conditioner system (PCS). (**b**) Converters topology employed in PCS.

Figure 2b shows the converter topology that is being used for this system. It can be seen that the hardware has the same bridge topology that can be used either as DC-DC, DC-AC, or AC-DC converter. However, at present the system is designed primarily for solar PV. By using this approach, the technology is made simple, which is important for application in remote islands. Modular plug and play feature may be expected of this system.

The power converters inside PCS are connected by an internal DC link bus (Figure 2b), whose voltage is being maintained at 500 V. At the beginning of operation, precharging action takes place as soon as the ESS is plugged into the storage chopper terminal. Once the DC link voltage is established, the management unit then activates the source chopper, inverter, and DC grid chopper depending on the specified SoC limits. Each PCS has the total capacity of 3 kW.

Table 1 lists the specification of the converters in every PCS, whereas Table 2 shows the state of each converter based on SoC. The flowchart in Figure 3 shows the decision-making process of the management unit according to guidelines set in Table 2.

**Table 1.** Specifications of each converter in a PCS unit.

|  | Source Chopper | Storage Chopper | DC Grid Chopper | Inverter |
|---|---|---|---|---|
| **Rated Power** | 3000 W | 3000 W | 1000 W | 3000 VA |
| **DC Link Voltage** | DC 500 V | DC 500 V | DC 500 V | DC 500 V |
| **Input/Output Voltage** | DC 250 V | DC 250 V | DC 370 V | AC 230 V |
| **Rated Current** | DC 12 A | DC 12 A | DC 2.7 A | AC 13 A |
| **Other** | MPPT Algorithm | Bidirectional power flow | Controlled DC Bus Voltage Range: DC 350–390 V Bidirectional power flow | Rated Frequency: 50 Hz (1-phase) |

**Table 2.** PCS converter states based on state of charge (SoC).

| SoC | Source Chopper * | Storage Chopper | DC Grid Chopper | Inverter |
|---|---|---|---|---|
| **0–10%** | ON or OFF | Charging | Charging | OFF |
| **10–30%** | ON or OFF | Charging | Charging | ON |
| **30–70%** | OFF | Charging | Charging | ON |
|  | ON | Discharging | Discharging |  |
| **70–100%** | ON or OFF | Discharging | Discharging | ON |

* Source chopper operates based on the availability of power generated by PV panels.

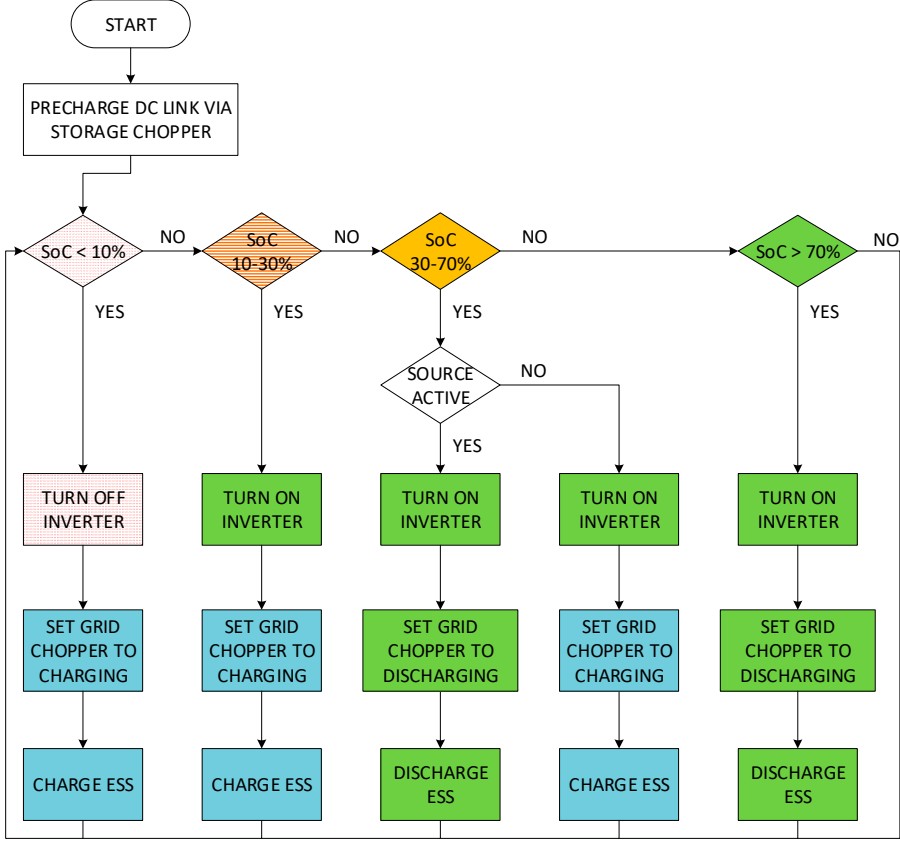

**Figure 3.** Flowchart of the PCS management unit.

The proposed DC microgrid system has the capability of working both as a single-point and multi-point system to serve single or multiple clusters of loads. Figure 4 shows the multi-point configuration with different kinds of loads. A single-point system converts power from the source and stores it in the battery and/or transfer it to the AC loads. In the multi-point mode, the proposed DC microgrid system works similar to a single point, but power sharing capability is through the DC bus. Each point controls its charge and discharge states with its own management unit the way it does in a single-point system and independent of what the other point does.

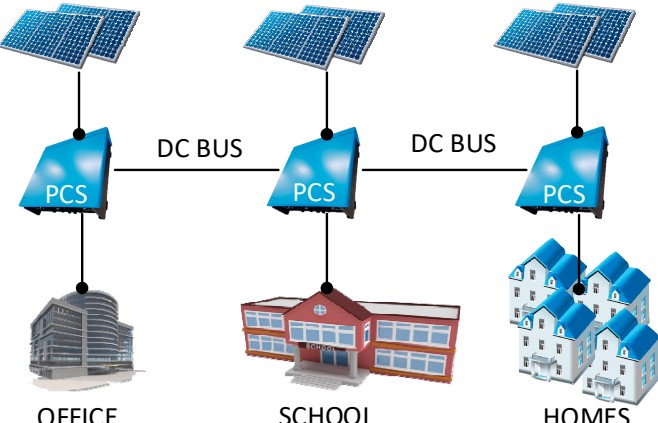

**Figure 4.** Multi-point DC microgrid configuration.

Figure 5 shows the power flow in the system based on the storage SoC level. The system that has higher energy level, or state of charge (SoC), exports power, whereas the system that lacks energy imports it. Consider the leftmost system. Its SoC is above 70%, so the DC grid chopper works in discharge mode regardless of what state the source chopper and storage chopper are in. Similarly, if the SoC is under 30%, then the management unit will automatically set the DC grid chopper to charging (not shown in the figure).

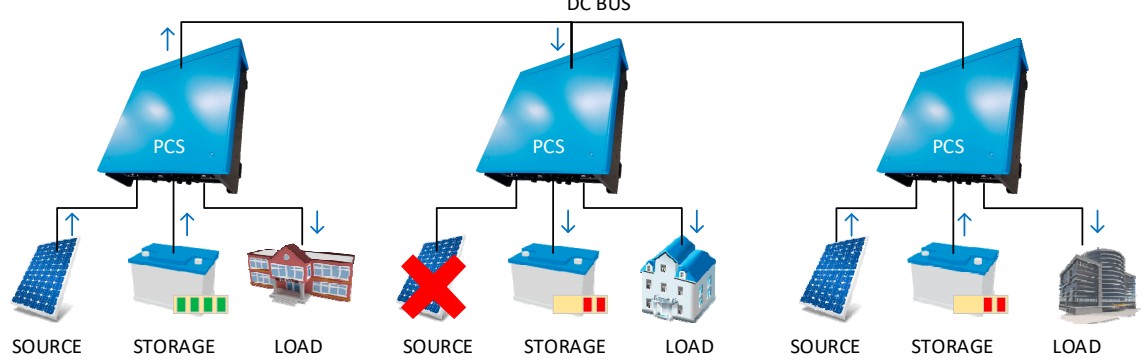

**Figure 5.** Power flow in multi-point DC microgrid system.

In the middle-level SoC, DC grid chopper operation is based on the status of the source and storage choppers. If the storage chopper is running, then power is supplied from the source and the PCS has the ability to send power out via the DC grid terminal. On the contrary, if the source chopper is not running, then the power of the unit is wholly supplied by battery. This causes the management unit to set the DC grid chopper to charging. Consider the middle and rightmost system in Figure 5. The system in the middle charges its storage system because the source is unavailable, while the system on the right-hand side exports power even though its storage is not in full condition because the source is available. The operational settings that are shown in Table 2 are configurable prior to field installation.

### 2.2. Implementation of the Proposed DC Microgrid System in Isolated Remote Areas

In rural areas, oftentimes the land to install public facilities is difficult to obtain because of unclear ownership or the land is forbidden to be used as per the local or cultural belief. To minimize the installation area, the proposed DC microgrid system can be configured as pole-mounted along with the other equipment such as PV panels and batteries. Figure 6 shows a solar tower with a set of PV panels, a PCS, batteries, and a control panel. Multi-point implementation of this configuration is shown in Figure 1.

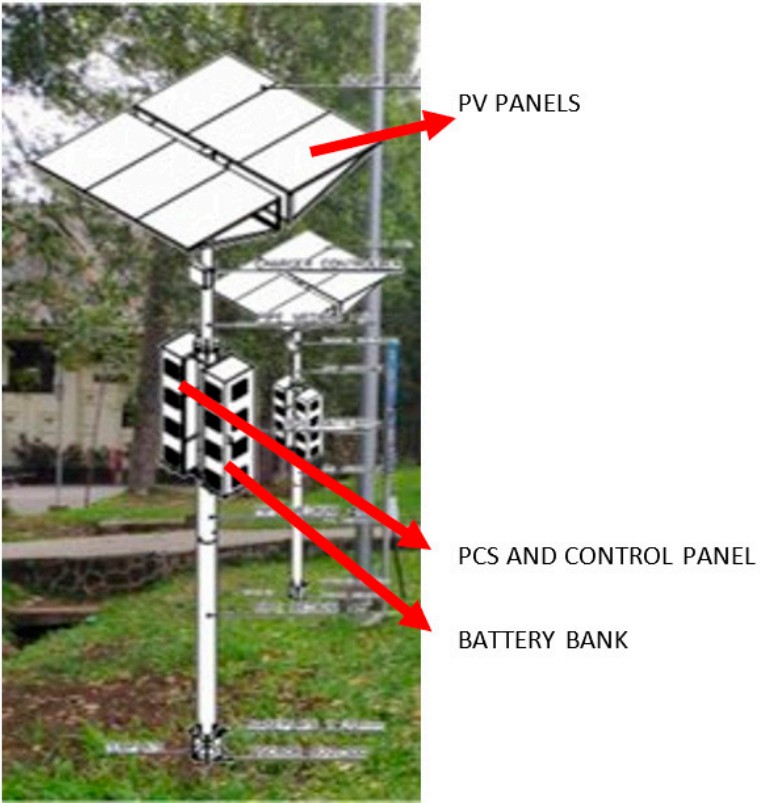

**Figure 6.** Solar tower: An implementation of PCS.

Houses in remote islands are assumed to use electricity for very basic needs, such as lighting, cooking, and, in places with access to communication facilities, to power up phones. Therefore, in this context, each house may be limited to use around 0.5 kWh each day. An individual system of the proposed DC microgrid has capacity of 3 kWp and 13.8 kWh of ESS per cluster, which is reasonable to serve 0.5 kWh per day for 10 houses. The system such as the one in Figure 6 uses 12 PV panels of 260 Wp each to achieve 3 kWp. The panels are positioned as a canopy shading around 18 m$^2$ of ground to minimize land utilization.

In the proposed system, every point is designed with higher power rating, where loads can connect directly to the load converter side of PCS. Table 3 compares the key characteristics of proposed method and several DC microgrid systems in the literature. All of these systems have modular components that make scaling up the size of the network easy, and each independent unit is designed for small-power use of rural homes. The main advantage of the proposed DC microgrid system over the others are the AC output into which the loads plug. Even though a lot of home appliances operate on DC power, AC-powered devices are still in abundance and easily obtained. By using a power system that outputs AC power, users are not required to change all of their electronic appliances. It will also be easier to switch to the AC utility grid when it reaches the remote areas.

**Table 3.** Comparison of several DC microgrid systems.

| No. | Source | Description | Topology |
|-----|--------|-------------|----------|
| 1. | Ref [12] | • Distributed generation that can be integrated with centralized power source and storage via DC grid.<br>• Every unit consists of PV panels, battery, and power converters.<br>• DC grid voltage: 380 V.<br>• Load voltage: 48 V.<br>• Power rating per unit: 125 W. |  |
| 2. | Ref [13] | • Centralized generation for several clusters of loads, one cluster consists of several households.<br>• Distributed voltage control assisted by digital communication.<br>• DC grid voltage: 360–400 V.<br>• Load voltage: 12 V.<br>• Power rating per unit: 100 W. |  |
| 3. | Ref [14] | • Distributed generation.<br>• DC grid voltage: 380 V.<br>• Load voltage: 48 V.<br>• Power rating per unit: 100 W.<br>• DC loads are connected directly to 48 V bus to avoid conversion losses. |  |
| 4. | Ref [15] | • Distributed generation.<br>• DC grid voltage: 380 V.<br>• Load voltage: 48 V.<br>• Power rating per unit: 200 W.<br>• Coordination between units is achieved by the converters' interaction within the units.<br>• Communal loads are supplied by the power generated from the home units. |  |

**Table 3.** *Cont.*

| No. | Source | Description | Topology |
|-----|--------|-------------|----------|
| 5 | Proposed System | <ul><li>Distributed generation.</li><li>DC grid voltage: 370 V.</li><li>Load voltage: 230 VAC 1-phase</li><li>Power rating per unit: 3 kW.</li><li>Each unit supplies a cluster of houses and manages its own energy consumption individually.</li><li>Outputs 230 VAC to accommodate common home appliances that run on AC power.</li></ul> | |

### 3. Functional and Performance Evaluation of the Proposed DC Microgrid System

*3.1. Evaluation Methodology*

There are two evaluation sections: Lab testing and field testing. In lab testing, the proposed DC microgrid system are tested as a single-point system and a multi-point system. Dummy SoC data are used to avoid having to charge or discharge the batteries to the desired level as doing so is too time-consuming. Single-point system evaluation mainly focuses on each point's functional specifications. Precharging sequence is done manually using 192-ohm resistor and 40-amps DC circuit breaker. The multi-point system evaluation tests the PCS operation sequence. Their DC grid terminals are being connected and the parameters observed. The units automatically precharge as they get connected to power source and then follow the same procedure as in single system initialization.

The parameters and equipment used for lab testing are as follow: Two sets of batteries: OPZV batteries 240-V 1,000-Ah and LiFePo batteries 256-V 70-Ah; two sets of PV modules: each consists of 40 50-Wp 12-V PV panels in series and parallel to achieve a total 340 V and 2 kWp; and two sets of 3-kW heater as resistive loads.

The field testing is conducted to gather data regarding conductor losses. Conductor losses need to be evaluated because in the target location for implementing this system, the residential area is thinly distributed. Figure 7 shows the field-testing setup. PCS are loaded with LED lights located 100 m away from it, assuming that the houses in rural villages could be this far apart from the energy source. Measurements are done at the AC output terminal of the PCS and at the load connection point. The charging and discharging process are also monitored with changing load currents.

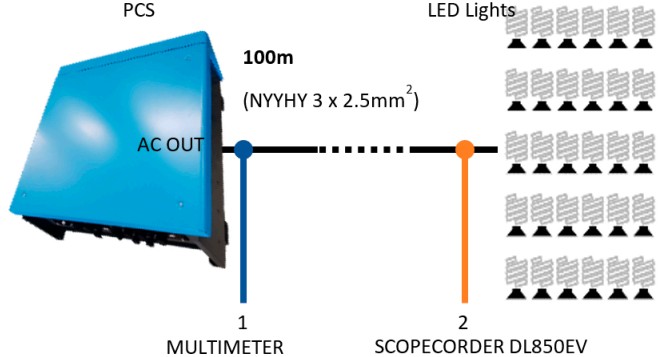

**Figure 7.** PCS field evaluation with distant loads.

### 3.2. Evaluation Results

### 3.2.1. Lab Testing

The storage chopper activates after the management unit (operated manually by using a laptop PC) gives the signal. Current drawn from battery indicates that the battery chopper is working. The measured voltage at DC link is 500 V. The battery chopper charges battery after the PV is connected to the source input terminal and the SoC in the management unit is set to 4%. The battery charging operation indicates that the PV chopper is working. When the inverter is activated and loaded, the current flows from both the PV and battery via their respective choppers.

Performance of the inverter is summarized in Figure 8. During this test, PV is deactivated, and the supply goes solely from the battery. The inverter voltage output is stable at 228 V AC, loaded at maximum 2.4 kW. The amount of DC power that is converted into AC is generally measured by the efficiency that can be calculated by using Equation (1).

$$Efficiency = \frac{AC\ Output}{Battery\ Input} \tag{1}$$

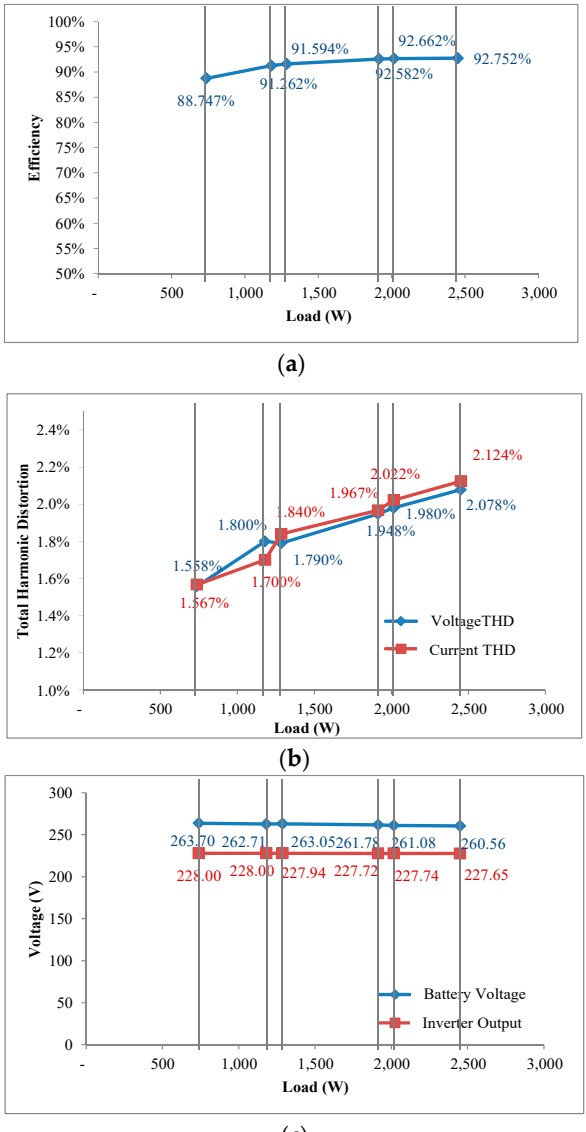

**Figure 8.** Inverter evaluation: (**a**) Efficiency vs. load, (**b**) voltage and current THD vs. load, and (**c**) battery and inverter voltage vs. load.

The efficiency data are shown in Figure 8a. The efficiency rises together with the load increase, reaching over 90% with load above 1100 W. It can be expected that at full load the efficiency will be higher than 92.75%. The power quality measurement results in Figure 8b shows that the harmonic level is well below standardized limits according to the IEEE Std 519-2014. The distortion levels increase along with the load in an almost linear manner. According to the trend, at full load they are predicted not to exceed the standard limit. The inverter also feeds stable voltage throughout the test, delivering up to 80% of rated output power without any significant drop in voltage, as shown in Figure 8c.

The DC grid chopper is being activated last. During this test, because the SoC is set to 4%, the DC grid chopper runs in charging mode (refer to Figure 4). When it is not connected to the other PCS to draw current from, the measured voltage at the DC grid chopper terminal is 390 V.

During this test, the power consumption of PCS units is also measured. In idle state when no switching occurs, the power consumed is 16–20 W. When switching occurs, the power consumption rises to 100 W.

For the multi-point evaluation, the precharging sequence is done automatically using precharging module within PCS. The precharging is done in 1.5 s and the discharging in 2 s. To make the PCS share power between each other, SoC of one PCS is set to 30% with the PV disconnected, while SoC of the other PCS is set to 70%. The management unit of the first PCS will detect that its battery needs to be charged. Therefore, management unit of the first PCS sets its DC grid chopper to charging mode. Meanwhile, management unit of the second PCS sets its DC grid chopper to discharging mode. Because the first PCS is set to absorb power from the DC bus and the second PCS to inject power, power will naturally flow from the second PCS to the first one.

### 3.2.2. Field Testing

Figure 9a shows that the inverter output voltage drops at the load point of connection. The proposed DC microgrid system does not use closed-loop control to keep constant voltage in the consumer to keep the simplicity. In addition, the household devices in remote area are not sensitive to constant voltage. At 9 A of load current (70% of rated current), the voltage drop does not exceed 10% of its nominal value, which is still acceptable for common AC-powered household devices. The voltage drops more when the load current is bigger due to the losses at the cable. Figure 9b shows the measured power at both end of the cable. During this experiment, the cable type NYYHY $3 \times 2.5$ mm$^2$ that is readily available in general stores was used. If a better quality is required, a different type of cable with less resistance may solve the problem.

Figure 10 shows the charging and discharging of battery bank with varying load current. This experiment is done by increasing and then decreasing the load demand. When the PV produces power more than the load demand (the charging process at the left-hand side), PCS will charge the battery, signified by the negative battery power. When the load demands power higher from the PV capacity (the discharging process in the middle), the battery enters discharging mode, hence the positive battery power. Battery gives out the necessary power to meet the load demand, and the power it exerts rises along with the increase in load current. PCS then goes back to charging mode when the load demand decreases to under the PV production. This experiment shows that the charging and discharging of the battery bank has worked well to meet the needs of the varying loads.

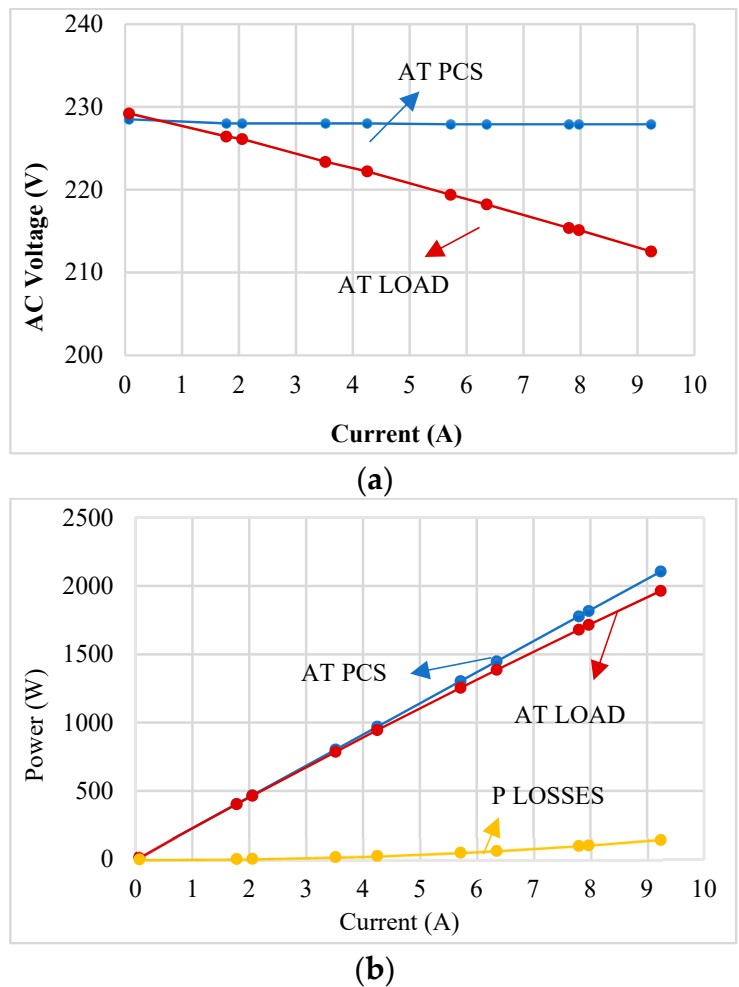

**Figure 9.** Field testing results: (**a**) Voltage drop on cable, and (**b**) losses on cable.

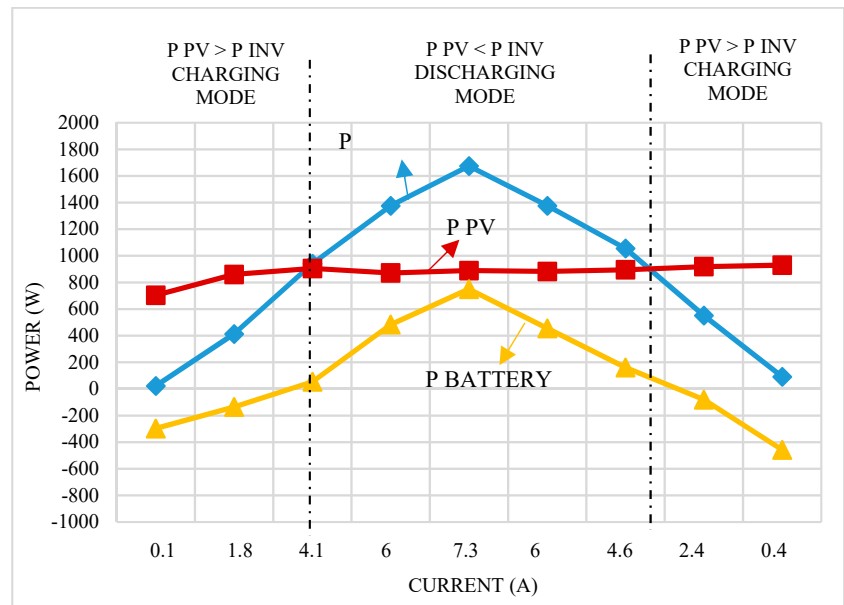

**Figure 10.** Battery charging and discharging process with varying load current.

### 4. LCOE and Investment Analysis

Levelized cost of energy (LCOE) is defined as the average total cost to build and operate a power plant per output energy in a certain period of time. LCOE can be calculated Equation (2).

$$LCOE = \frac{NPVCost}{NPVProduce} = \frac{\sum \frac{I(t)+M(t)+F(t)}{(1+r)^t}}{\sum \frac{E(t)}{(1+r)^t}} \tag{2}$$

where:

$I$ = initial investment cost,
$M$ = O&M cost,
$F$ = fuel cost,
$E$ = energy produced by the plant,
$r$ = discount rate,
$t$ = time,
*NPVCost* = net present value from the total cost spent in the plant's lifetime, and
*NPVProduce* = net present value of the total energy generated by the plant during its lifetime.

The *LCOE* is computed using the following assumptions: Three systems such as described in Figure 1 will be installed, which amount to total capacity of 9 kWp, each system can be used to serve 10 homes running common home electronic appliances; investment cost of each system is US$23,375; system efficiency is 95%; sun light is accessible for an average 4.5 h/day all year; PV panel output degradation is 0.5%/year; O&M cost is 0.5% from investment cost/year; O&M growth rate is 2%; interest rate is 9%/year, and because 70% of the investment comes from bank loan, the discount rate is 10.8%; and the project lifespan is 20 years.

For 20 years, the total *NPV* for investment and O&M is US$72,975 and the total generated energy is 107.4 kW. Therefore, the $LCOE = \frac{NPVCost}{NPVProduce} = \frac{US\$72,975}{107.4} = US\$0.68/kWh$. If the tariff is the same as *LCOE*, the internal rate of return (IRR) is 10.82% and pay out in 8.26 years. This implies that in order for this project to be economically feasible, the *LCOE* specifies the minimum tariff for this system. US$ 0.68/kWh is inexpensive compared to diesel generator that is generally the main electricity source in remote areas. Adding up the costs of transporting fuels, the total production cost may be well above US$ 1/kWh. Moreover, by using the proposed DC microgrid, we will reduce dependency on the already dwindling fossil fuel reserves.

### 5. Conclusions

This paper proposes a new DC microgrid system concept that is suitable for accelerating electricity delivery to rural and remote areas that has no access to the main utility grid. Its modularity makes it easily transported and provides flexibility to the overall system. The DC interface between points is also advantageous when they need to combine with other points due to the absence of the need to synchronize as in AC-interfaced systems.

PCS is the key equipment for the proposed microgrid system. It interfaces the microgrid elements such as energy sources, ESS, and load, by using the same converter topologies for those different purposes. It can be deployed as an individual isolated unit and also form a grid through the DC bus for better reliability and efficiency. PCS manages the available power and the load requirements, while able to exchange power between connected points.

Prototypes have been developed to demonstrate the proposed DC microgrid system, with capacity of 3 kWp of solar PV and 13.8 kWh of battery each. One such system can be used to supply a cluster of 10 households. They are implemented in a pole-mounted structure to save space, while the PV panels form a canopy shading 18 m$^2$ of ground. The system can be expanded to create a DC microgrid network by connecting several single isolated systems through 370 VDC bus.

The proposed DC microgrid system has been tested in laboratory and in the field, both as a single-point and a multi-point system. Each of them runs well both in either modes, reaching 92.75% of efficiency, and confirmed low THD levels, where the voltage THD is 2.078% and current THD is 2.124%. The prototype field testing successfully demonstrated the battery charge and discharge sequences following several load conditions. Voltage drops is also tested to confirm the acceptable level of voltage in the consumer's household devices. Financial analysis based on LCOE concludes that the proposed system is feasible for implementation with the minimum tariff US$0.68/kWh.

**Author Contributions:** Conceptual development P.A.D., A.R., and T.A.; economic analysis E.G.; implementation and field testing A.H.A., P.D.S., and M.R. All authors have read and agreed to the published version of the manuscript.

**Funding:** This research is funded by the LPDP, Indonesia. This research is also partially funded by the Indonesian Ministry of Research and Technology/National Agency for Research and Innovation, and Indonesian Ministry of Education and Culture, under World Class University Program managed by Institut Teknologi Bandung.

**Institutional Review Board Statement:** Not applicable.

**Informed Consent Statement:** Not applicable.

**Data Availability Statement:** Not applicable.

**Acknowledgments:** The authors would also like to thank Sanken Electric Co., Ltd. and PT Len Industri (Persero), for cooperation during this research.

**Conflicts of Interest:** The authors declare no conflict of interest.

**Abbreviations**

| | |
|---|---|
| AC | Alternating Current |
| DC | Direct Current |
| GWp | gigawatt-peak |
| ESS | Energy Storage System |
| kWh | kilowatt-hour |
| kWp | kilowatt-peak |
| LCOE | Levelized Cost of Energy |
| O&M | Operation and Maintenance |
| PCS | Power Conditioner System |
| PV | Photovoltaic |
| RES | Renewable Energy Sources |
| SoC | State of Charge |
| THD | Total Harmonic Distortion |

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
