# Peer review of "A DC Microgrid System for Powering Remote Areas†"

_energies, doi:10.3390/en14020493_

Round 1

Reviewer 1 Report

This paper proposes an integrated generation-storage unit for dc microgrids.

The proposed unit has some interesting aspects, but I would like to ask some clarifications:

  • Please expand a bit the literature review in the introduction. Several papers are cited at line 42 but there is no critical analysis of their content. Can you please specify a bit more into detail what is the content of these papers? The reader otherwise can find it difficult to understand which paper is more relevant.
  • Figure 2: can you please provide more detail on the topology of the converters? For example, I guess that the DC/AC inverter is a single phase one, but it should be mentioned explicitly. Also, I think that arrows can be added in figure 2 to explain the direction of the power flow (which converter is unidirectional and which is bidirectional)
  • Table 1 states that the maximum dc link voltage is 450 V, however, at line 166 you state that the measured dc link voltage is 500 V. Can you please explain this better? Maybe you could also add the rated voltages in figure 2.
  • Solar tower: what is the area covered by the pv panels?
  • I do not really see the point of figs.9 and 10. I think that this test does not prove the effectiveness of the proposed PCS, but simply demonstrates that a 100m long cable has losses.
  • Figure 11: I think this figure is a bit confusing. Why is the x axis simmetrical? Is there a hidden dependance from time?
  • At line 247 you say that each system can serve 10 homes. Can you please add the typical electrical consumption of the considered homes? For example a typical European household features a consuption larger than 3 kW, therefore it can be a bit confusing to see such numbers.

Moreover, I think the authors should expand a bit their claims in the introduction. They can stress more the advantages of the proposed solution in terms of compactness and limited cost.

The quality of the text must be improved. There are several grammar mistakes and some typos (eg, "DC microgrid system system" at line 69). Please correct them.

Reviewer 2 Report

Thank you for your contribution. Your work can enhance the quality of life of people with limited access to electricity. Congrats on your achievement! 

1- Figure 1 is not clear and the text within is not readable.

2- Would you explain what is the difference between the proposed method and rooftop solar PV microgrid?

3- In section 3.2.1 you mentioned lab testing. Could you explain in details the set of equipment used in this experiment. For example, the storage chopper switching frequency, and electrical ratings. Similarly, with the inverter. More details is better. 

4- Add a flowchart for control scheme if possible.

5- You successfully completed your experiment and implementation. However, you have not compared your method to any other method. I suggest you make a table and compare results, advantages, and disadvantages. This way the readers will understand the significance of your contribution.  

6- Add more references. Recent references are preferred. 

7- Add a nomenclature section.

8- Some figures might violate the page margins. you might need to resize some figures, such as figure 2. 

Reviewer 3 Report

The authors provided interesting and meaningful work, but the reviewer had some questions:

  • The objectives need to be more clearly defined.
  • The main advantages and disadvantages of the present study must be added
  • There exist some parts should be improved, such that point (Related work) isn't shown in this manuscript. Please carefully check and revise the whole manuscript.
  • The conclusions must be improved. The conclusions should include the results obtained in the test
  • The references must be added.

Reviewer 4 Report

The paper discuss the remote control of power in dc microgrid system. However, the following points need to answer and revise paper. 1) From the power rating in the paper, the microgrid should be changed to "nanogrid" from the point of power rating. 2) All of the power converters are not discussed in the paper. Because the power converter topologies are very important to make sure the dc voltage stability on the dc grid voltage at 400V. 3) I think the paper did not present any control scheme to control the power flow and ........... 4) The paper quality is not a technical paper quality.

Round 2

Reviewer 1 Report

Thank you for your replies and the modifications to the text.

Author Response

Thank you

Author Response

thank you

Reviewer 3 Report

More references should be added.

The abstract should be improved, with more information and more important objectives developed. 

The conclusions should include more information about the results obtained.

Reviewer 4 Report

The authors have answered some questions in detail.

Author Response

thank you